# Sex Differences in the Load–Velocity Profiles of Three Different Row Exercises

**DOI:** 10.3390/sports11110220

**Published:** 2023-11-09

**Authors:** Raúl Nieto-Acevedo, Blanca Romero-Moraleda, Almudena Montalvo-Pérez, Carlos García-Sánchez, Moisés Marquina-Nieto, Daniel Mon-López

**Affiliations:** 1Departamento de Deportes, Facultad de Ciencias de la Actividad Física y del Deporte, Universidad Politécnica de Madrid, Calle Martín Fierro, 7, 28040 Madrid, Spain; nietoacevedoraul@gmail.com (R.N.-A.); c.gsanchez@upm.es (C.G.-S.); moises.mnieto@upm.es (M.M.-N.); daniel.mon@upm.es (D.M.-L.); 2Department of Physical Education, Sport and Human Movement, Autonomous University of Madrid, 28049 Madrid, Spain; blanca.romero@uam.es; 3Applied Biomechanics and Sports Technology Research Group, Autonomous University of Madrid, 28049 Madrid, Spain; 4Faculty of Sport Sciences, Universidad Europea de Madrid, 28670 Madrid, Spain

**Keywords:** velocity-based training, gender, mean propulsive velocity

## Abstract

This study examined the force–velocity profile differences between men and women in three variations of row exercises. Twenty-eight participants (14 men and 14 women) underwent maximum dynamic strength assessments in the free prone bench row (PBR), bent-over barbell row (BBOR), and Smith machine bent-over row (SMBOR) in a randomized order. Subjects performed a progressive loading test from 30 to 100% of 1-RM (repetition maximum), and the mean propulsive velocity was measured in all attempts. Linear regression analyses were conducted to establish the relationships between the different measures of bar velocity and % 1-RM. The ANOVAs applied to the mean velocity achieved in each % 1-RM tested revealed significantly higher velocity values for loads < 65% 1-RM in SMBOR compared to BBOR (*p* < 0.05) and higher velocities for loads < 90% 1-RM in SMBOR compared to PBR (*p* < 0.05) for both sexes. Furthermore, men provided significantly higher velocity values than women (PBR 55–100% 1-RM; BBOR and SMBOR < 85% 1-RM; *p* < 0.05) and significant differences were found between exercises and sex for 30–40% 1-RM. These results confirm that men have higher velocities at different relative loads (i.e., % 1-RM) compared to women during upper-body rowing exercises.

## 1. Introduction

Strength training is fundamental for enhancing muscular strength, size, and power [1]. It is well known that maximizing resistance training effects depends on the manipulation of program variables such as intensity, volume, rest interval, duration, etc. Among them, intensity is one of the key variables in training and, commonly, it is identified with relative load (percentage of one-repetition maximum [% 1-RM]) [1]. Several studies have shown that the traditional 1-RM test is valid and reliable in determining the muscular strength of athletes [2,3]. The traditional direct 1-RM assessment uses an incremental loading test to directly assess the 1-RM. The standard procedure involves performing a specific exercise with increasing loads until reaching the maximum lifting capacity, considering the test complete when the individual can no longer perform a successful repetition with a higher load [4]. However, the traditional 1-RM test could have several limitations, such as being time-consuming to assess each athlete, which makes its use more complicated in big groups of individuals (i.e., team sports) [5,6]. Additionally, the risk of injury can be high because this measurement is performed under maximum loading conditions [5,6]. Also, other factors such as training fatigue, nutrient intake, and sleep can alter the 1-RM daily test with fluctuations around 36% [7,8].

The use of technology in sport can provide information for optimizing the prescription and monitoring of training programs [9]. Furthermore, technology has enabled the development of new training methodologies such as “velocity-based resistance training” to address the limitations of the traditional 1-RM test [9]. Bar velocity measurement has been proposed as an alternative method to estimate accurately the 1-RM [6,10]. This method is called velocity-based training (VBT), which is a reliable indirect method based on the load–velocity relationship (L–V) to predict 1-RM in resistance training [10,11]. Several studies have examined the relationship between the percentages of % 1-RM and the corresponding mean propulsive velocity (MPV) [6,10]. The L–V relationship improves the conventional procedure to calculate the 1-RM by using the velocity output to apply a regression model. The incremental test starts with light load until reaching the 1-RM. The fastest repetition performed with each load by each subject is used for the regression model. Once the 1-RM load is known, the linear regression model is applied to the experimental points provided by all subjects to establish the generalized relationship between % 1-RM and lifting velocity. Generalized L–V relationships have been established for a variety of RT exercises, including the squat [12,13], deadlift [14,15,16,17,18], hip-thrust [19,20], leg press [13], leg extension [16], bench press [6,21], bench pull [22,23], military press [17,24], and pull-up [5,18]. The generalized L–V relationships allow to determine what is the % 1-RM that is being used as soon as the first repetition with a given load is performed with maximal voluntary velocity in the exercises studied [6].

Most VBT studies have focused on two main movements: the upper-body bench press exercise and the lower-body squat [25]. While these exercises are important for many sporting activities, other movements, such as pull exercises like the prone row, are crucial for conditioning athletes in various sports (e.g., swimming, climbing, rowing, judo) [26,27,28,29]. There are different variations of the pull exercises, which are a multijoint upper-body exercises that can be performed by athletes and nonathletes for improving strength of the posterior shoulder girdle, back and elbow flexor muscles [23,30,31,32,33], and upper-body pulling [23,30,31] and pushing power [34]. It has been commonly prescribed in “periodized” strength and conditioning programs for elite rowers, kayakers, sailors, swimmers, and rugby league players [23,30,31,33]. It also has been used as an assessment of one repetition maximum (1-RM) upper-body pulling strength and power [31,33,34,35]. Although rowing exercises may have important applications in sports performance, not much research has been conducted on velocity-based training [25]. Moreover, most studies have been conducted using Smith machines [36]. During the free-weight back squat, typical errors of 0.03–0.05 m/s^−1^ across loads of 20–90% of 1-RM have been shown [36]. Therefore, we consider it important to include in our study the same exercise but in Smith machine and free-weight conditions. Moreover, many VBT studies have been conducted in men, showing that men generally present higher velocity values for light to moderate loads compared to women at the same relative loads (% 1-RM). These findings have been reported across various exercises, including the squat, deadlift, hip thrust, inclined bench press, seated chest press, and seated military press [4,19,21,24,37,38].

To address the existing gaps in the literature, further examination of the load–velocity relationship during row exercises is required. Then, the purpose of this research was to compare the load–velocity profile between men and women during bent-over barbell row (BOBR), free prone bench row (FPBR), and bent-over Smith machine row (BOSMR). We hypothesized that (I) the men would present higher velocities at each % 1-RM than women in the three rows studied and (II) the velocity attained at each relative load (% 1-RM) would be similar in the three exercises. It is crucial to know the response of each sex to the load–velocity relationship to prescribe more accurately the training programs for both sexes. The results of this research not only will inform appropriate loading prescriptions for effective training based on sex, but also will serve as a valuable tool for evaluating progress, setting goals, and establishing benchmarks in various athletic and research contexts.

## 2. Materials and Methods

### 2.1. Participants

We conducted a sample size estimate using GPower 3.1 Software (Dusseldorf, Germany), based on the effects of similar loads on 1-RM and mean test velocity (MTV) output observed from the flat bench press exercise [21]. For an alpha level of 0.05 and power of 0.90, sample sizes from 3 (for 1-RM) to 6 (for MTV) per group appeared to be necessary to detect the significant loading effects of [39]. We conservatively recruited twenty-eight participants, 14 men (age = 25.1 ± 6.0 years; body mass = 82.5 ± 10.9 kg; height = 181.2 ± 5.37 cm) and 14 women (age = 24.2 ± 5.2 years; body mass = 61.6 ± 6.6 kg; height = 164.8 ± 6.7 cm) who volunteered to participate in this study. Inclusion criteria were: (1) having at least 2 years of resistance training experience; (2) not having any health or musculoskeletal injuries that could compromise testing; and (3) having training experience in the exercises tested. After being informed of the purpose and testing procedures, subjects signed a written informed consent form before participation. The present investigation was approved by the Research Ethics Committee of the Universidad Politécnica de Madrid and was conducted following the Declaration of Helsinki [40].

### 2.2. Experimental Design

A counterbalanced-measures design was used. Subjects were tested in random order for the individual load–velocity relationship and the determination of the 1-RM strength using a 20 kg barbell (Eleiko, Halmstad, Sweden) in the three modes of rowing. Participants attended the laboratory four times (one day for a familiarization session and another three days to test each day one exercise) (Figure 1), with an interval of approximately 48–72 h between visits to reduce interferences caused by consecutive maximal assessments (e.g., neuromuscular fatigue or post-activation potentiation effects [41]). During each visit, they were submitted to a standardized 1-RM test in the bent-over barbell row (Figure 2A), bent-over Smith machine row (Figure 2B), and prone bench row (Figure 2C) (Hammer-strength, Rosemont, IL, USA) in a randomized order (Research Randomizer, www.randomizer.org). The three rows were performed with a prone grip and hands were separated by a distance equivalent to the participant’s acromion-to-acromion length. To consider a repetition valid, the participant needed to start the movement with the elbows fully extended and end the movement when the bar contacts the abdomen or the bench. For the bent-over barbell row and bent-over Smith machine row, the legs are bent slightly, and the upper body is also bent until it is almost perpendicular to the floor, keeping the spine in a natural/neutral position. Finally, the barbell rises towards the lower part of your chest. For the prone bench row, the torso remains rigid and motionless on the bench. The head and the neck are aligned with the trunk by tucking the chin inward slightly and resting it gently on or slightly off the edge of the padded bench surface [42]. On the three occasions, participants were requested to avoid strenuous exercises and beverages containing caffeine/alcohol for 24 h before testing. The testing sessions were carried out at the same place and time of day (±1 h) for each subject, under the same environmental conditions.

### 2.3. Testing Procedures

The warm-up protocol consisted of three minutes of stationary cycling at a self-selected easy intensity and five minutes of joint mobilization exercises, followed by six repetitions with fixed loads of 30 and 20 kg for men and women, respectively. The load–velocity test followed standard procedures [22]. The initial load was set at 20 kg and gradually increased in increments of 5–10% 1-RM until an MPV of 0.8 m·s^−1^ was reached, performing three repetitions with each load. Two repetitions were performed when the MPV was between 0.8 and 0.6 m·s^−1^, and only one repetition for higher loads. The heaviest load that each subject could properly lift while completing a full range of movement and without any external help was considered to be their 1-RM. Inter-set rests were 3 min to reduce possible neural or mechanical fatigue [37]. Only the best repetition (fastest and performed correctly) on each load was considered for subsequent analysis. All repetitions were recorded with a linear velocity transducer (Speed4lift, Madrid, Spain) [43] that was attached to the left side of the barbell and recorded vertical velocity at a frequency of 1000 Hz. The MPV–mean velocity value was considered in the present study. MPV is the mean velocity value from the start of the concentric phase until the acceleration of the barbell is lower than gravity (−9.81 m/s^−2^) [44]. Strong verbal encouragement was provided during all tests to motivate participants to give maximal effort.

### 2.4. Statistical Analysis

Data are presented as means (M), standard deviation (SD), and Pearson’s correlation coefficient (r). The normal distribution of the data was confirmed by Shapiro–Wilk, and the homogeneity of variances was confirmed by Levene’s test (*p* > 0.05). Pearson’s correlation coefficients were used to quantify the association of the 1-RM value and the velocity of 1-RM (V1-RM) between the three exercises. An ANOVA was applied to each dependent variable (i.e., mean velocity values attained at each % 1-RM, 1-RM strength) with the exercise (BBOR, PBR, and SMBOR) as a within-participant factor and sex (men and women) as a between-participant factor. When significant differences were observed, a Bonferroni’s post hoc comparison was performed. Therefore, the Mann–Whitney U test for independent samples was used to test between-group differences for the age, height, and body mass. Significance was set at *p* < 0.05. The Pearson’s correlation was interpreted based on the recommendations of Schober et al. [45], where ≤0.10 represents negligible correlation, 0.10–0.39 is a weak correlation, 0.40–0.69 is a moderate correlation, 0.70–0.89 is a strong correlation, and ≥0.90 is a very strong correlation. Results analyses were carried out using a custom spreadsheet (Microsoft Excel version 16.69.1) and JASP software version 0.16.4 (Nieuwe Achtergracht, Amsterdam).

## 3. Results

The load–velocity profile of the three upper-body rowing exercises was well fitted by a linear regression model when the data of all participants of the same sex were pooled (Figure 3). The slope of the generalized participants’ load–velocity profiles was always steeper in men than in women. A linear regression model also fitted the sex load–velocity profiles for BBOR (men: $r^{2}$ = 0.928; women: $r^{2}$ = 0.777), SMBOR (men: $r^{2}$ = 0.934; women: $r^{2}$ = 0.834), and PBR (men: $r^{2}$ = 0.825; women: $r^{2}$ = 0.771).

The ANOVAs applied on the mean velocity achieved in each % 1-RM tested revealed significantly higher velocity values for loads <65% 1-RM in SMBOR compared to BBOR in women (*p*-range < 0.001–0.004). Significantly higher velocity values for loads from 70 to 100% 1-RM in SMBOR compared to PBR were found in women and from 30 to 90% 1-RM in men (*p*-range < 0.001–0.05). Significantly higher velocity values for load 70% 1-RM in BBOR compared to PBR in men and from 85 to 100% 1-RM in women (*p* < 0.001) (Table 1) were also identified.

Furthermore, men provided significantly higher velocity values than women against light and medium loads (PBR 55–100% 1-RM; BBOR < 85% 1-RM; and SMBOR < 80% 1-RM) (*p*-range < 0.001–0.05), and significant exercise × sex interactions were observed from 30 to 40% 1-RM (*p*-range < 0.032–0.046) (Table 1).

There were significant differences for the anthropometric measures (height and BM; *p* < 0.001), although no significant differences were found for age (*p* = 0.917). Men presented a higher value of 1-RM strength (Table 2) and ratio-scaled 1-RM (Figure 4) than women for all exercises (*p* < 0.001).

## 4. Discussion

This study explored the load–velocity profile during three upper-body rowing exercises (BBOR, PBR, and SMBOR) in men and women. Supporting our first hypothesis, men provided significantly higher velocity values than women against light and medium loads in BBOR and SMBOR and against moderate to high in PBR. Contrary to our second hypothesis, SMBOR showed significantly higher velocity values for the loads compared with the PBR, especially in women with loads lower than 65% 1-RM. Furthermore, men presented a higher 1-RM than women for any rowing exercise. Significant exercise × sex interactions were observed from 30 to 40% 1-RM. Finally, men showed more r^2^ than women for all exercises (men: r^2^ range = 0.825–0.934 vs. women: r^2^ range = 0.771–0.834).

The differences in favor of men in BBOR and SMBOR with light and medium loads are in line with the findings of previous studies. Alonso-Aubin et al. [46] observed that women obtained lower values for maximal velocity at 40% of 1-RM in adolescent rugby players [46]. In the same line, women attained inferior velocities (<85% 1-RM) in horizontal and incline bench press and seated military press exercises [21,47]. Furthermore, Pareja Blanco et al. [37] found that men produce significantly higher mean propulsive velocities from 30 to 90% 1-RM, practically across the whole spectrum of loads [37]. These results also support our findings in PBR, where the differences between sexes were in high loads. An alternative reason to clarify the mechanisms of the differences in the load–velocity profiles between men and women is that men seem to have more fast muscle fibers [48]. This could explain why individuals with a high proportion of fast muscle fibers show better performance in high-velocity movements or actions [49].

Another interesting finding was that the variance between exercises found higher velocity values for the Smith machine bent-over row. Previous studies have reported that the load–velocity profile is exercise-specific [23,44,50]. Although the differences in these studies were in the V1-RM of the compared exercises, we did not observe differences in the V1-RM between the three variations of the row. One possible explanation for this is that the exercises have the same movement pattern. Similar results were obtained in the study by Amador García et al. [21] which analyzed three upper-body pushing exercises [21]. However, our study also showed differences for the Smith machine bent-over row compared with the two other rows in moderate and low loads. This fact may be due to the greater stabilization of the Smith machine in bent-over row exercise and, therefore, a better ability to apply force. Consequently, an unstable condition decreases the parameters of strength, power, and muscular speed [51].

For that comparison of 1-RM strength, a ratio scaling of the 1-RM performance normalized to the BM was used to diminish the large differences in BM between men and women [52,53]. In this study, men had higher absolute (Table 2) and relative (Figure 4) 1-RM performance than women for BBOR, PBR, and SMBOR. These results are consistent with previous research reporting higher neuromuscular performance in men compared to women [52,53,54]. For example, Jones et al. [53] found that men produced a higher absolute average (50% difference) and peak power (42% difference) in a range of loads in the deadlift exercise (*p* < 0.001). Similarly, Komi et al. [54] reported lower muscular power (31.7%; *p* < 0.001), total leg force (19.7%; *p* < 0.001), quadriceps force (29.0%; *p* < 0.001), and force–time performance (50.3%; *p* < 0.001), along with other physiological differences (i.e., muscle enzyme activities, electromyographic activity, muscle fiber composition, etc.), in a group of young women compared to their male counterparts. Furthermore, although long-term training has been shown to minimize sex-related differences in neuromuscular function, the higher lean BM of men would ultimately promote their superior strength capacity [53].

In addition, we obtained strong linear relationships between the averaged MPV across the subjects and relative loads for the three rows. These data are in line with other studies showing that *r*^2^ for the load–velocity relationship obtained during upper- [21,24] and lower-limb exercises [13,37] revealed strong correlations (≥0.70). Moreover, our analyses indicate that the strength of the linearity of the relationship between velocity (MPV) and relative load (% 1-RM) is slightly lower in free-weight rows (BBOR and PBR) compared with the Smith machine row (SMBOR). In this vein, these results provide, for García Ramos [4], support that the % 1-RM-velocity relationship is equipment-specific, showing differences during the bench press when it is performed using a Smith machine compared to using free weights. Interestingly, it was also observed that we obtained lower *r*^2^ values for women in the three exercises than for men. This lower *r*^2^ value may also be explained by higher levels of body mass in men, but a larger sample size would be necessary to confirm this statement. It has also been proposed that the determination of the individual load–velocity relationship overcomes these limitations by improving the quality of estimations [4]. These findings suggest the importance of calibrating the individualized L–V relationship based on the sex and equipment used in the exercise.

Although we emphasized the perpendicular alignments (concerning the ground) to measure vertical displacements accurately in free exercises, the precision of the velocity measurements could be more accurate in the Smith machine because it limits the horizontal oscillations of the barbell. Another limitation could be the heterogeneity of the participants’ strength status, though the differences between men and women are not directly caused by their different strength levels [47].

Nevertheless, this study provides additional support for the necessity of using women’s formulas instead of general (based on men’s) equations for women, because our results confirm that men and women attain different velocities against each % 1-RM, particularly with light and moderate loads. From a practical standpoint, strength and conditioning coaches could determine resistance training intensity and control the strength performance of their women athletes quickly and efficiently in BOBR, FPBR, and SMBOR in many sports where pull movement is important. Future studies should attempt to compare the precision of both sexes (men and women) load–velocity relationships to predict the 1-RM during different resistance training exercises and different populations (e.g., basketball players vs. football players).

## 5. Conclusions

In conclusion, women have lower velocity values than men during the barbell and Smith machine bent-over exercises, particularly with loads under 85% 1-RM, and lower velocities in free prone bench row for loads > 60% 1-RM. The bent-over Smith machine row has shown to be the exercise with higher velocities. Consequently, it will be necessary to use the specific equation provided for each sex if we want to estimate the load from velocity with high precision. Men reported significantly higher ratio-scaled 1-RM performance than women in all exercises (BOBR, FPBR, and SMBOR). Our findings highlight the importance of integrating a well-designed strength and conditioning program based on the sex of the athlete. Research on women has gained popularity over the past two decades; studies such as this one allow sport practitioners to determine resistance training intensity and control strength fast and efficiently. With the information provided here, men and women who use velocity-based training in their practices will be able to include a new set of upper-body pull movements (i.e., free prone bench pull, bent-over barbell row, and bent-over Smith machine row) individualized with accuracy through velocity.

## Figures and Tables

**Figure 1 sports-11-00220-f001:**
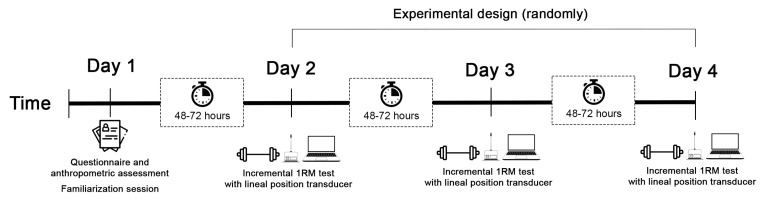
Study timeline.

**Figure 2 sports-11-00220-f002:**
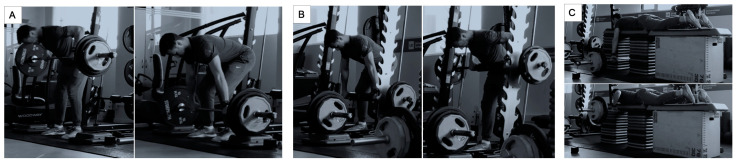
Bent-over barbell row (**A**), bent-over Smith machine row (**B**), and prone bench row (**C**).

**Figure 3 sports-11-00220-f003:**
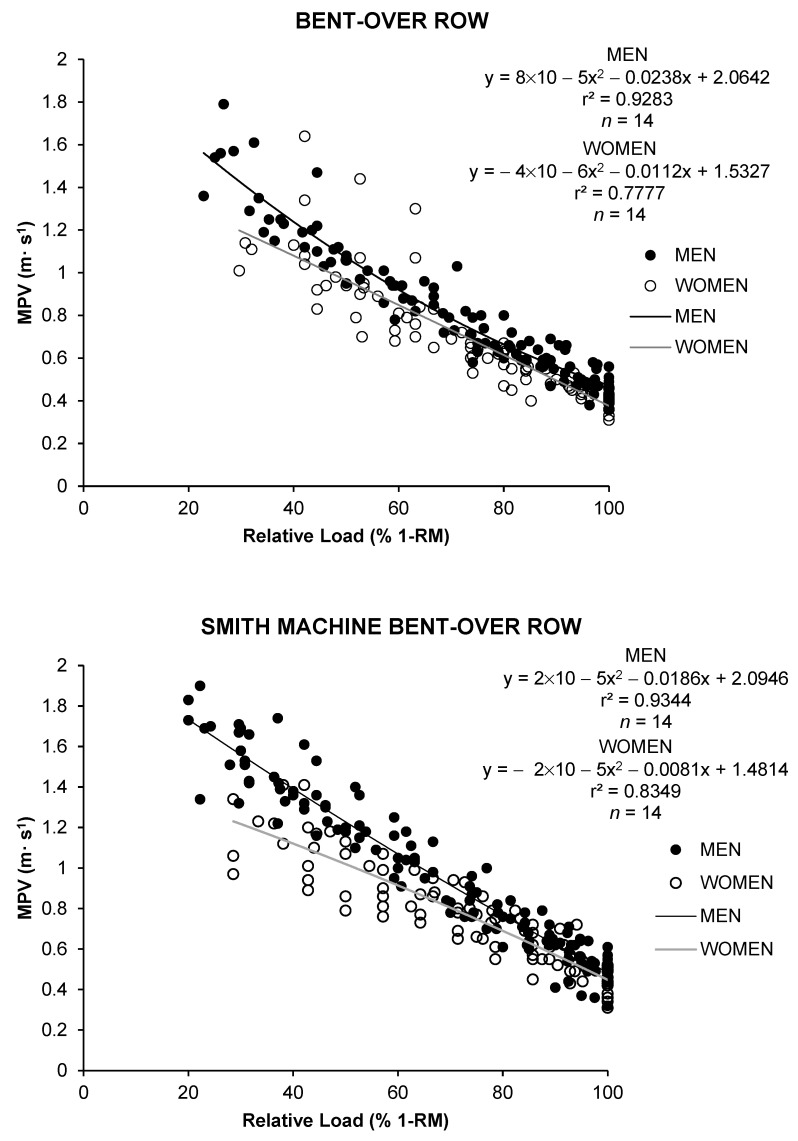
Load–velocity profiles were obtained separately from the bent-over row (**upper panel**), the Smith machine bent-over row (**middle panel**), and the prone bench row (**lower panel**) (data averaged across the participants). The linear regression models are shown separately for men (straight line and filled circles) and women (dashed line and open circles). *r*^2^, Pearson’s multivariate coefficient of determination; *n*, number of trials included in the regression analysis.

**Figure 4 sports-11-00220-f004:**
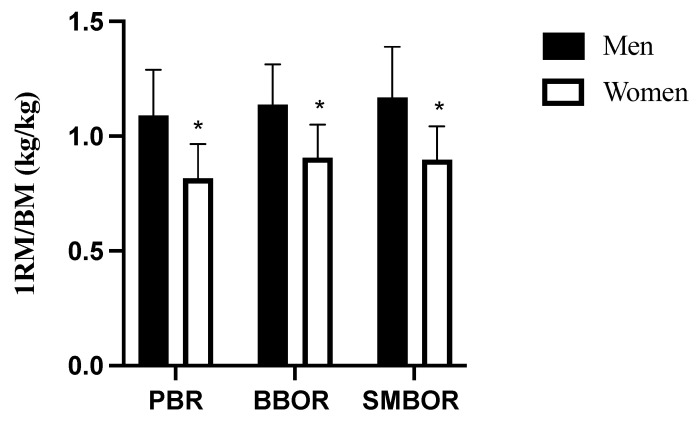
Comparison of relative 1-RM performances between sexes (mean ± SD). * Shows significantly (*p* < 0.001) lower than men group. 1-RM = one repetition maximum; BM = body mass; PBR = prone bench row; BBOR = barbell bent-over row; SMBOR = Smith machine bent-over row.

**Table 1 sports-11-00220-t001:** Comparison of the MPV attained against each relative load between the three row exercises in men and women.

Load % 1-RM	ANOVA (*p*-Value)			Prone Bench Row (m/s)		Barbell Bent-OVER Row (m/s)		Smith Machine Bent-Over Row (m/s)	
				Men	Women	Men	Women	Men	Women
	Row	Sex	Row × Sex	Mean ± SD	Mean ± SD	Mean ± SD	Mean ± SD	Mean ± SD	Mean ± SD
30%	0.004	<0.001	0.032	1.35 ± 0.15	1.21 ± 0.27	1.44 ± 0.20	1.10 ± 0.19 *	1.55 ± 0.12 ^#^	1.24 ± 0.23 *^†^
35%	0.003	<0.001	0.038	1.29 ± 0.14	1.15 ± 0.25	1.37 ± 0.19	1.06 ± 0.17 *	1.47 ± 0.11 ^#^	1.19 ± 0.21 *^†^
40%	0.003	<0.001	0.046	1.22 ± 0.13	1.10 ± 0.23	1.30 ± 0.17	1.02 ± 0.16 *	1.39 ± 0.10 ^#^	1.13 ± 0.19 *^†^
45%	0.002	<0.001	0.058	1.16 ± 0.12	1.03 ± 0.21	1.24 ± 0.16	0.97 ± 0.14 *	1.32 ± 0.10 ^#^	1.08 ± 0.17 *^†^
50%	0.001	<0.001	0.078	1.09 ± 0.11	0.97 ± 0.18	1.17 ± 0.15	0.93 ± 0.13 *	1.24 ± 0.09 ^#^	1.03 ± 0.15 *^†^
55%	0.001	<0.001	0.111	1.02 ± 0.10	0.91 ± 0.16 *	1.11 ± 0.14	0.89 ± 0.12 *	1.17 ± 0.08 ^#^	0.97 ± 0.13 *^†^
60%	<0.001	<0.001	0.167	0.96 ± 0.10	0.85 ± 0.14 *	1.04 ± 0.12	0.85 ± 0.10 *	1.09 ± 0.08 ^#^	0.92 ± 0.12 *^†^
65%	<0.001	<0.001	0.268	0.89 ± 0.09	0.79 ± 0.12 *	0.98 ± 0.12	0.80 ± 0.10 *	1.01 ± 0.07 ^#^	0.86 ± 0.10 *^†^
70%	<0.001	<0.001	0.436	0.83 ± 0.08	0.73 ± 0.10 *	0.91 ± 0.11 ^#^	0.76 ± 0.09 *	0.94 ± 0.07 ^#^	0.81 ± 0.09 *^#^
75%	<0.001	<0.001	0.669	0.76 ± 0.08	0.67 ± 0.08 *	0.84 ± 0.10	0.72 ± 0.08 *	0.86 ± 0.07 ^#^	0.76 ± 0.09 *^#^
80%	<0.001	0.001	0.815	0.70 ± 0.07	0.61 ± 0.06 *	0.78 ± 0.10	0.68 ± 0.08 *	0.79 ± 0.07 ^#^	0.70 ± 0.09 *^#^
85%	<0.001	0.003	0.791	0.63 ± 0.07	0.55 ± 0.05 *	0.71 ± 0.10	0.63 ± 0.09 *^#^	0.71 ± 0.07 ^#^	0.65 ± 0.09 ^#^
90%	<0.001	0.024	0.573	0.57 ± 0.06	0.49 ± 0.04 *	0.65 ± 0.11	0.59 ± 0.09 ^#^	0.64 ± 0.07 ^#^	0.59 ± 0.10 ^#^
95%	<0.001	0.128	0.358	0.50 ± 0.06	0.43 ± 0.04 *	0.58 ± 0.11	0.55 ± 0.10 ^#^	0.56 ± 0.08	0.54 ± 0.11 ^#^
100%	<0.001	0.404	0.215	0.44 ± 0.06	0.37 ± 0.05 *	0.52 ± 0.12	0.51 ± 0.11 ^#^	0.48 ± 0.08	0.49 ± 0.13 ^#^

1-RM, one-repetition maximum; ANOVA, analysis of variance; in bold, significant *p*-value; *, significant differences between sexes (*p* < 0.05); ^#^, significant differences from prone bench row (*p* < 0.05); ^†^, significant differences from barbell bent-over row (*p* < 0.05).

**Table 2 sports-11-00220-t002:** One-repetition maximum (1-RM), body mass (BM), and ratio-scaled 1-RM (1-RM/BM) between the three row exercises in men and women.

	Prone Bench Row		Barbell Bent-Over Row		Smith Machine Bent-Over Row	
	Men	Women	Men	Women	Men	Women
	Mean ± SD	Mean ± SD	Mean ± SD	Mean ± SD	Mean ± SD	Mean ± SD
1 RM (kg)	90.00 ± 19.12 *	50.39 ± 11.49	93.65 ± 17.16 *	55.962 ± 11.25	99.04 ± 23.20 *	55.39 ± 11.45
BM (kg)	82.5 ± 10.9 *	61.6 ± 6.6	82.5 ± 10.9	61.6 ± 6.6	82.5 ± 10.9	61.6 ± 6.6
1 RM/BM (kg)	1.09 ± 0.20 *	0.817 ± 0.148	1.14 ± 0.18 *	0.91 ± 0.14	1.17 ± 0.22 *	0.90 ± 0.15

1-RM, one-repetition maximum; BM, body mass; kg, kilograms; MPV, mean propulsive velocity. * shows significant differences between sexes (*p* < 0.05).

## Data Availability

Not applicable.

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
