# Peer review of "Sex Differences in the Load–Velocity Profiles of Three Different Row Exercises"

_sports, 2023, doi:10.3390/sports11110220_

Round 1
Reviewer 1 Report
Comments and Suggestions for Authors
Firstly, I would like to thank the editor and the authors for the opportunity to review this manuscript. Secondly, I must say that my comments are exclusively of a scientific nature, aimed at helping you understand the work and its scientific merit.
Although the paper is very well written, I regret to say that in my eyes the originality of this study is very incipient. In this context, I believe that the potential for adding knowledge is unfortunately limited. I suggest the authors revisit their results to try to extract more information from what was produced in the study.
There are formatting errors, such as punctuation before references (pg. 2; line 58), please proofread the entire text.
Why didn't you measure 1MR in the traditional way for each exercise to compare it to that predicted by VBT?
The fact of including exercises that have not yet been studied, or even comparisons between sexes, does not necessarily make the research original. So I would ask the authors what new information their article brings, and what addition to the body of knowledge of physical education and sport they intend to make with this publication.
Although the names of the exercises are similar, from a mechanical point of view I only see similarities between the bent-over barbell row and the bent-over Smith-machine row, while the prone bench row is a totally different exercise that requires the muscle groups involved differently from the two previous exercises. So, what is the point of evaluating the three exercises and comparing them?
The non-parametric relationships between strength and speed variables have been established for a long time, as has the fact that men, due to their muscular composition, are on average more capable than women of performing muscular activities that involve a great deal of strength. Based on this consideration, please clarify what innovation is inherent in the study.
Author Response
Dear reviewer,
We would like to thank you for taking the necessary time and effort to review the manuscript. We sincerely appreciate all your valuable comments and suggestions, which helped us improve the quality of the manuscript.
There are formatting errors, such as punctuation before references (pg. 2; line 58), please proofread the entire text.
Thank you for your comment. We have checked the manuscript.
Why didn't you measure 1MR in the traditional way for each exercise to compare it to that predicted by VBT?
Thank you for your comment. We followed the protocol of most research on this topic to analyze the load-velocity profile (1–7). We performed an incremental test until 1RM and recorded the Mean Propulsive Velocity (MPV) associated with this 1RM (we have added a table with the 1RM tested in our study; pg. 8; line 7; Table 2). After that a linear regression model is applied to
establish the individualized %1RM-MPV relationship. In this way, it is likely to provide the most accurate individualized %1RM-MPV relationship (8).
The fact of including exercises that have not yet been studied, or even comparisons between sexes, does not necessarily make the research original. So I would ask the authors what new information their article brings, and what addition to the body of knowledge of physical education and sport they intend to make with this publication.
This exercises have been studied (5,9), but just in men. Thus, our intention was to compare both sexes to analyze if sex differences existed in MPV. We think that our results could have a significant impact on athletic performance where small difference has a big influence on the competition (e.g., in Olympic shooter (10), elite sprinters (11), in Competitive Powerlifters (12). In addition, there are many studies that show the importance of individualizing training according to gender due to there are significant differences due to psychology (13), menstrual cycle (14–17) or anthropometric differences (18).
Although the names of the exercises are similar, from a mechanical point of view I only see similarities between the bent-over barbell row and the bent-over Smith-machine row, while the prone bench row is a totally different exercise that requires the muscle groups involved differently from the two previous exercises. So, what is the point of evaluating the three exercises and comparing them?
As we mentioned before there are a previous study analyzing these three exercises but only for men participants (9). We think that these three exercises share muscle mass involved in because the three exercises are horizontal pull where there are elbow flexion and shoulder extension (19). Moreover, Kulig et al., (20) suggest that the pull exercises are a descending-strength curve exercise where maximum strength is produced at the start of the lift. In addition, in this study (9), the authors found that the optimum power loads in the three exercises occur at fixed ranges of bar-velocities. So, we compared the three exercises because with our results could individualize training for each exercise based on the sex and it may facilitate the work of strength and conditioning specialists interested in sports where upper body pull exercises play a crucial role (21–24).
The non-parametric relationships between strength and speed variables have been established for a long time, as has the fact that men, due to their muscular composition, are on average more capable than women of performing muscular activities that involve a great deal of strength. Based on this consideration, please clarify what innovation is inherent in the study.
We used a parametric test (ANOVA repeated measures) because most of studies have used this statistical test (1–7). We agree with your comment, it is well-established that the muscular fiber of men and women are different (25). However we considered that is necessary to know if these differences in muscular composition are related with the different manifestations of strength and how sex impacts on the load–velocity relationship. Some studies have shown differences in the mean propulsive velocity at which men and women get each %1RM (7,26–30). Therefore, if we train based on velocity and we prescribe the same velocity for men and women, they would be training at different relative intensities (%1RM).
References:
1. González-Badillo JJ, Sánchez-Medina L. Movement velocity as a measure of loading intensity in resistance training. Int J Sports Med. 2010 May;31(5):347–52.
2. García-Ramos A, Pestana-Melero FL, Pérez-Castilla A, Rojas FJ, Haff GG. Differences in the load–velocity profile between 4 bench-press variants. Int J Sports Physiol Perform. 2018 Mar 1;13(3):326–31.
3. Jukic I, García-Ramos A, Malecek J, Omcirk D, Tufano JJ. The Use of Lifting Straps Alters the Entire Load-Velocity Profile During the Deadlift Exercise. J strength Cond Res. 2020 Dec;34(12):3331–7.
4. Lopes dos Santos M, Mann B, Berton R, Alvar B, Lockie R, Dawes J. Using the Load-Velocity Profile for Predicting the 1RM of the Hexagonal Barbell Deadlift Exercise. J Strength Cond Res. 2022 Jan 12;Publish Ah.
5. García-Ramos A, Ulloa-Díaz D, Barboza-González P, Rodríguez-Perea Á, Martínez-García D, Quidel-Catrilelbún M, et al. Assessment of the load-velocity profile in the free-weight prone bench pull exercise through different velocity variables and regression models. PLoS One. 2019;14(2):e0212085.
6. Rodiles-Guerrero L, Pareja-Blanco F, León-Prados JA. Comparison of load-velocity relationships in two bench press variations: weight stack machine vs Smith machine. Sport Biomech. 2020;00(00):1–13.
7. Pareja-Blanco F, Walker S, Häkkinen K. Validity of using velocity to estimate intensity in resistance exercises in men and women. Int J Sports Med. 2020 Dec;41(14):1047–55.
8. García Ramos A. Resistance training intensity prescription methods based on lifting velocity monitoring. Int J Sports Med. 2023 Aug 22;
9. Loturco I, Suchomel T, Kobal R, Arruda A, Guerriero A, Pereira L, et al. Force-Velocity relationship in three different variations of prone row exercises. J Strength Cond Res. 2018 Feb 15;33.
10. Moreira da Silva F, Malico Sousa P, Pinheiro VB, López-Torres O, Refoyo Roman I, Mon-López D. Which are the most determinant psychological factors in olympic shooting performance? A self-perspective from elite shooters. Int J Environ Res Public Health. 2021 May 1;18(9).
11. Haugen T, Seiler S, Sandbakk Ø, Tønnessen E. The Training and Development of Elite Sprint Performance: an Integration of Scientific and Best Practice Literature. Sport Med - Open 2019 51. 2019 Nov 21;5(1):1–16.
12. Dugdale JH, Hunter AM, Di Virgilio TG, Macgregor LJ, Hamilton DL. Influence of the “slingshot” bench press training aid on bench press kinematics and neuromuscular activity in competitive powerlifters. J Strength Cond Res. 2019 Feb 1;33(2):327–36.
13. Nuzzo JL. Narrative Rreview of sex differences in muscle strength, endurance, activation,size, fiber type, and strength training participation rates, preferences, motivations,injuries and neuromuscular adaptations. J Strength Cond Res. 2023 Feb;37(2):494–536.
14. Hunter SK. The relevance of sex differences in performance fatigability. Med Sci Sports Exerc. 2016 Nov;48(11):2247–56.
15. Elliott-Sale KJ, McNulty KL, Ansdell P, Goodall S, Hicks KM, Thomas K, et al. The Effects of Oral Contraceptives on Exercise Performance in Women: A Systematic Review and Meta-analysis. Sports Med. 2020 Oct;50(10):1785–812.
16. Carmichael MA, Thomson RL, Moran LJ, Wycherley TP. The Impact of Menstrual Cycle Phase on Athletes’ Performance: A Narrative Review. Int J Environ Res Public Health. 2021 Feb 9;18(4):1667.
17. Meignié A, Duclos M, Carling C, Orhant E, Provost P, Toussaint J-F, et al. The Effects of Menstrual Cycle Phase on Elite Athlete Performance: A Critical and Systematic Review. Front Physiol. 2021;12:654585.
18. Fragala MS, Clark MH, Walsh SJ, Kleppinger A, Judge JO, Kuchel GA, et al. Gender Differences in Anthropometric Predictors of Physical Performance in Older Adults. Gend Med. 2012 Dec;9(6):445.
19. Lorenzetti S, Dayer R, Plüss M, List R. Pulling Exercises for Strength Training and Rehabilitation: Movements and Loading Conditions. J Funct Morphol Kinesiol 2017, Vol 2, Page 33. 2017 Sep 19;2(3):33.
20. Kulig K, Andrews JG, Hay JG. Human strength curves. Exerc Sport Sci Rev. 1984;12(1):417–66.
21. Barbado D, Lopez-Valenciano A, Juan-Recio C, Montero-Carretero C, Van Dieën JH, Vera-Garcia FJ. Trunk stability, trunk strength and sport performance level in judo. PLoS One. 2016 May 1;11(5).
22. Pearson SN, Cronin JB, Hume PA, Slyfield D. Kinematics and kinetics of the bench-press and bench-pull exercises in a strength-trained sporting population. http://dx.doi.org/101080/14763140903229484. 2009;8(3):245–54.
23. Aspenes ST, Karlsen T. Exercise-training intervention studies in competitive swimming.
24. Vigouroux L, Devise M, Cartier T, Aubert C, Berton E. Performing pull-ups with small climbing holds influences grip and biomechanical arm action. J Sports Sci. 2019 Apr 18;37(8):886–94.
25. Nuzzo JL. Sex differences in skeletal muscle fiber types: A meta-analysis. Clin Anat. 2023 Jul 10;
26. Fitas A, Santos P, Gomes M, Pezarat-Correia P, Mendonca G V. Influence of sex and strength differences on the load–velocity relationship of the Smith-machine back squat. Sport Sci Health. 2023;
27. García-Ramos A, Suzovic D, Pérez-Castilla A. The load-velocity profiles of three upper-body pushing exercises in men and women. Sport Biomech. 2021 Jul;20(6):693–705.
28. Balsalobre-Fernández C, García Ramos A, Jimenez-Reyes P. Load–velocity profiling in the military press exercise: Effects of gender and training. Int J Sports Sci Coach. 2017 Oct 26;13(5):743–750.
29. Nieto-Acevedo R, Romero-Moraleda B, Montalvo-Pérez A, Valdés-Álvarez A, García-Sánchez C, Mon-López D. Should We Use the Men Load-Velocity Profile for Women in Deadlift and Hip Thrust? Int J Environ Res Public Health. 2023 Mar 1;20(6).
30. Nieto-Acevedo R, Romero-Moraleda B, Díaz-Lara FJ, Rubia A de la, González-García J, Mon-López D. A Systematic Review and Meta-Analysis of the Differences in Mean Propulsive Velocity between Men and Women in Different Exercises. Sport 2023, Vol 11, Page 118. 2023 Jun 13;11(6):118.
Reviewer 2 Report
Comments and Suggestions for Authors
It is well written study. I have only few suggestions before the publication.
L1-3- remove dot from the title.
L16 – “the force-velocity…” add profiles.
L22-24- this statement is general for both groups? Please Clarify.
Introduction – is slightly superficial, but also informative. I would like to only suggest adding after the last paragraph statement about application of your results.
L77 – due to inclusion criteria, did you gather a data about any other physical activity?
Figure 1 should be in better resolution.
In conclusion you repeated your results description. Please add a brief explanation ( you did it well in the discussion) and underline the practical application of your results.
Author Response
Dear reviewer,
We would like to thank you for taking the necessary time and effort to review the manuscript. We sincerely appreciate all your valuable comments and suggestions, which helped us improve the quality of the manuscript.
L1-3- remove dot from the title. Thank you for your comment. We have checked.
L16 – “the force-velocity…” add profiles. Thank you for your comment. We have added.
L22-24- this statement is general for both groups? Please Clarify.
Thank you for your comment. Right. this statement is for both sexes, we have specified it in line 24.
Introduction – is slightly superficial, but also informative. I would like to only suggest adding after the last paragraph statement about application of your results.
Thank you for your suggestion. We have added a possible application about our results. Line 81-85.
L77 – due to inclusion criteria, did you gather a data about any other physical activity?
Thank you for your comment. In this study, we just required that participants have at least 2 years of experience, especially in the exercises tested. Without any doubt, it would be interesting to have more data about the sample (e.g., type of sport, frequency of practice, number of hours per week…)
Figure 1 should be in better resolution. We have changed because it was impossible to get a better resolution.
In conclusion, you repeated your results description. Please add a brief explanation ( you did it well in the discussion) and underline the practical application of your results.
Thank you for your suggestion. We have clarified in conclusion and underlined the practical applications. Line 249-255.
Reviewer 3 Report
Comments and Suggestions for Authors
Dear Sports Editors and Authors, my comments about the manuscript are as follow, thank you by this inviting.
The review of the article is positive, highlighting the importance of strength training, the limitations of the traditional 1-RM test, and the need for a velocity-based approach. The research objectives are clear, aiming to compare load-velocity profiles between men and women in various rowing exercises.
The methods are well described and adequately address participant recruitment, experimental design, testing procedures, and statistical analysis. The use of safety measures, such as intervals between testing sessions, is appropriate to minimize interference. The use of standardized protocols and suitable equipment is also a solid approach. In summary, the methods are rigorous and suitable for addressing the proposed research questions.
The results provide clear evidence that men generally achieve higher velocities and 1-RM strength than women in upper-body rowing exercises. The linear regression models applied to load-velocity profiles demonstrate a consistent difference between the sexes, with men exhibiting steeper slopes, indicating greater velocity increases with increasing load. These findings suggest that men and women may benefit from different training strategies and loading prescriptions in rowing exercises.
The study's approach to analyzing load-velocity profiles and comparing these profiles between sexes and exercises is sound and provides valuable insights into strength and velocity differences. However, it would be beneficial to explore the practical implications of these findings in terms of training recommendations for both men and women. Additionally, further research could delve into the underlying physiological mechanisms contributing to these differences and their impact on athletic performance.
In other way, the findings revealed that men generally achieved higher velocity values compared to women, particularly with various load ranges in different rowing exercises. These differences align with previous research suggesting that men tend to exhibit superior performance in high-velocity movements. The study also highlighted exercise-specific variations in velocity values, with the Smith-machine bent-over row showing higher velocities. While there were limitations related to measurement precision and participant heterogeneity, the study emphasized the importance of using sex-specific equations to accurately estimate training intensity based on velocity. Overall, the findings underscore the need for tailored strength and conditioning programs based on an athlete's sex.
Overall, the results contribute to our understanding of the load-velocity relationship in upper-body rowing exercises and highlight the importance of considering sex-specific factors in strength training programs.
Author Response
Dear reviewer,
We would like to thank you for taking the necessary time and effort to review the manuscript. We sincerely appreciate all your valuable comments and suggestions, which helped us improve the quality of the manuscript.
The study's approach to analyzing load-velocity profiles and comparing these profiles between sexes and exercises is sound and provides valuable insights into strength and velocity differences. However, it would be beneficial to explore the practical implications of these findings in terms of training recommendations for both men and women. Additionally, further research could delve into the underlying physiological mechanisms contributing to these differences and their impact on athletic performance.
Thank you for your comment. We have tried to clarify the practical implications of the conclusion. Line 249-255
Round 2
Reviewer 1 Report
Comments and Suggestions for Authors
The authors modified the wording of the study, which facilitated its understanding and contributions to the field. In this context, I believe that the questions raised have been adequately addressed, allowing the paper to be accepted.